# Fatigue Strength Enhancement of Butt Welds by Means of Shot Peening and Clean Blasting

**Jonas Hensel \*** , **Hamdollah Eslami, Thomas Nitschke-Pagel and Klaus Dilger**

Institute of Joining and Welding, Technische Universität Braunschweig, Langer Kamp 8,
38106 Braunschweig, Germany

\* Correspondence: j.hensel@tu-braunschweig.de; Tel.: +49-531-391-95517

**Abstract:** Shot peening is a mechanical surface treatment to improve the fatigue strength of metallic components. Similarities exist between regular shot peening and conventional industrial clean blasting. However, the main difference between these two processes is the peening media used and the lack of control and documentation of peening parameters. The clean blasting process is not yet qualified to optimize fatigue enhancement, although it holds a similar potential to regular shot peening. Clean blasting is frequently applied to welded components, with the purpose of surface preparation for application of corrosion protection. This article presents the results of regular shot peened double V-groove (DV) butt welds made from construction steels S355N and S960QL, as well as the high strength aluminum alloy Al-6082. The peening parameters are varied widely. Furthermore, the effect of coverage and intensity is investigated to test the robustness of the peening processes. The data is completed with industrially clean blasted welds, representing typical workshop conditions. The overall objective of this work is to derive minimum peening parameters that still allow significant fatigue strength benefits. The presented data show a high robustness of the fatigue results to peening parameters.

**Keywords:** welded joints; fatigue strength; steel; aluminum; post-weld treatment; shot peening; clean blasting; residual stresses

---

## 1. Introduction

### 1.1. Post-Weld Treatment of Welds

Post-weld treatment of welds for fatigue strength enhancement has become common in many industry applications. Grinding and thermal stress annealing, in particular, are widely used and accepted by technical standards, such as [1,2]. In the past few years, mechanical surface treatment methods have become more and more popular in the welding industry. Recently, the most discussed has been high frequency mechanical impact hammer peening (HFMI), which is locally applied to fatigue critical weld toes. Mechanical surface treatment methods, such as HFMI, utilize different fatigue beneficial effects to some degree [3,4]. These effects are the generation of compressive residual stress, cold work hardening, and geometric changes of the weld profile. The International Institute of Welding (IIW) has already adopted HFMI treatment in terms of a widely accepted guideline [5]. Further examples of accepted post-weld treatments are Tungsten Inert Gas (TIG)-dressing, as well as conventional hammer and needle peening [6]. However, all these mentioned methods have in common is that their application usually results in additional production costs.

This article focusses on shot peening and clean blasting as alternative mechanical surface treatments. The main difference between these two processes is that shot peening is applied aiming at fatigue strength enhancements, mainly applied in industrial engineering, while clean blasting aims

generally at surface cleaning. Furthermore, the surface roughness can be adjusted by clean blasting for the application of corrosion protection. Thus, shot peening is used under control of the peening parameters (e.g., shot media, peening intensity, coverage), while clean blasting is mainly controlled by the appearance of the surface (e.g., cleanliness, roughness). The peening media used for clean blasting is normally of lower quality (multiple re-use) and edged. Nevertheless, clean blasting has similar effects as shot peening in terms of the generation of compressive residual stress and cold work hardening.

Shot peening is widely used in the automotive and aviation industries and has proven its potential for fatigue strength enhancement. A good overview on the principles and applications of shot peening can for instance be found in [7,8]. Unfortunately, the benefits of shot peening on fatigue strength cannot yet be used in the design of welded structures, because shot peening is not currently covered by design codes. This article presents fatigue test results from shot peened and clean blasted DV-butt welds. The objective of this work is to investigate the effects of different peening media and peening parameters on fatigue results of different construction metals. Finally, it should be possible to qualify regularly applied clean blasting as a post-weld treatment method. Hopefully, this would result in a way to increase the fatigue resistance of welded components without significantly increasing manufacturing costs.

### 1.2. Shot Peening as a Post-Weld Treatment Method of Welded Joints

Shot peening is a flexible peening method for surfaces of various metallic and non-metallic components. Typical applications are the enhancement of surface roughness, cleanliness, or the enhancement of mechanical parameters and corrosion resistance [3]. The most important peening parameters of shot peening, in terms of fatigue improvement, are shot size, shot geometry, peening intensity, and coverage [9]. There are several shot peening media available, such as steel, glass, ceramic, or organic components. The user can influence the kinematic energy of the peening impacts by the choice of the peening media, shot size, impact angle, and shot velocity. Steel components are normally shot peened using steel balls. Glass beads are used in the case of demands for low peening intensity, for instance, low sheet thickness. Furthermore, they are used to smooth surfaces (finishing) or to avoid (chemical) reactions between peening media and peened components. Clean blasting is normally performed using cheaper peening media, like sand, corundum, or broken cast steel. The possibility for re-use of the peening media depends on the requirements regarding the surface quality of the peened component. The shot peening process can be further adjusted with the help of the peening parameters, which are controlled by the Almen test [10].

The peening intensity is a measure for the kinematic energy that is transferred from the shots to the surface. In practice, this is controlled by the empirical Almen strip test. The Almen strip is a steel specimen with defined thickness, hardness, and mechanical parameters. Almen strips are peened using a defined parameter set, resulting in a specific plastic deformation of the strip. The intensity of the peening process is measured by the plastic deformation of the Almen strips and denoted in mm (or inch) and the Almen strip used. Coverage of the surface is defined as the percentage of the peened surface related to the un-peened surface. A coverage of 98% is the highest coverage value that can be experimentally determined, as the indentation spots of the shots are still distinguishable here. A coverage of 98% is normally denoted as "full" coverage. Higher or lower values of coverage are normally adjusted by a control of the peening time per surface area, e.g., 0.25 × 98% means a reduction of the peening time to 25%. Shots should ideally hit the surface at 90° impact angle.

Two effects are relevant for the fatigue improvement of shot peening: (1) the generation of compressive residual stress, and (2) cold work hardening. The reduction of surface roughness by plastic deformation may be of advantage in cases of high strength metals (hardened steels). The induced compressive residual stress field may retard fatigue micro cracks or slow down crack growth considerably. The magnitude of the induced compressive residual stress and the penetration depth of the residual stress field are important. The effects of residual stress on the fatigue strength of peened

surfaces depend on the ultimate strength and can be expressed by means of the sensitivity to residual stress *m* [11].

Cold work hardening can be utilized in un-notched or mildly notched specimens to enhance their fatigue strength. The fatigue strength of such specimens depends on the ultimate strength of the material. The significance of this effect is more pronounced in metals with low hardness. Hard metals show less sensitivity to cold working. Furthermore, an increase in roughness lowers the effectiveness of beneficial fatigue effects through cold work hardening in the case of high hardness [3]. Investigations on fatigue strength improvements of welded steels by shot peening can be found in the literature, for instance [12–16]. The general success of shot peening is proven well. However, the peening parameters were not varied widely. The fatigue strength improvement of welded aluminum alloys by shot peening has also been proven already [17]. Peening with steel shots and standard peening parameters (0.2 mmA, 200% coverage) was applied to metal inert gas (MIG)- and TIG-welded Al-5083 alloys. The fatigue strength of welds was improved almost up to the base metal fatigue strength. However, the possibility of softening in the heat affected zone must always be considered in cases of high strength precipitation hardening alloys.

## 2. Experimental Work

The effect of shot peening and clean blasting on the fatigue strength of general metal arc (GMA)-welded butt welds was investigated using three different materials, steel grades S355N (1.0545) and S960 (1.8933), as well as aluminum Al-6082 T6 (EN AW-AlSi1MgMn/3.2315). The sheet thickness was 10 mm (steel) and 5 mm (aluminum). The peening media were varied, reflecting regular shot blasting (steel shots S280) as well as clean blasting (glass beads and corundum). Furthermore, the coverage (peening time) was varied between $0.25 \times 98\%$ (low coverage) and $2 \times 98\%$ (double coverage), simulating different scenarios. A low coverage of 25% reflects the lower boundary of clean blasting of a time-optimized cleaning process, while $2 \times 98\%$ coverage is used in conventional shot peening processes. Another parameter investigated was the influence of the peening intensity. This parameter was described by means of the Almen intensity, measured as the deflection of specific Almen strips of varying thickness, Table 3. The three peening intensities used here were 0.3 to 0.4 mmA, mmN, and mmC (0.3 to 0.4 mm deflection of Almen strips A, N and C respectively). An overview is given in Tables 1–3. Specimens prepared by means of combinations of the aforementioned parameters were used for fatigue testing. The surface roughness, hardness, surface, and near surface residual stresses were documented.

**Table 1.** Base metals used for specimen preparation: mechanical and chemical parameters determined by tensile test respectively spectroscopy.

| Material | Yield Limit | Tensile Strength | Elongation at Fracture | CEV | Thickness |
|---|---|---|---|---|---|
| S355N | 376 MPa | 518 MPa | 27% | 0.41 | 10 mm |
| S960QL | 995 MPa | 1033 MPa | 16% | 0.56 | 10 mm |
| Al-6082 T6 | 317 MPa | 340 MPa | 17% | - | 5 mm |

**Table 2.** Peening media used.

| Parameter | Steel Shots S280 | Glass Beads | Corundum NK 24 |
|---|---|---|---|
| Particle size | 600–1180 μm | 300–400 μm | 595–841 μm |
| Shape | spherical | spherical | edged |
| Density | 7.3–7.8 g/cm$^3$ | 2.6 g/cm$^3$ | 3.9 g/cm$^3$ |
| Hardness | 400–520 HV | 500–530 HV [1] | 8–9 [1] (Mohs hardness) |

[1] Hardness taken from the literature [3].

**Table 3.** Almen strips used.

| Almen Strip | N | A | C |
|---|---|---|---|
| Thickness | 0.79 ± 0.02 mm | 1.29 ± 0.02 mm | 2.39 ± 0.02 mm |

## 2.1. Specimen Preparation and Characterization

The specimens used for fatigue testing were prepared as DV-butt welds (60° opening angle). The power source was elmatech DV36 L(W) by elmatech, Germany. Plates of approximately 400 mm length were attached to each other by means of a robot-guided standard impulse MAG (metal active gas welding) or MIG (metal inert gas welding) process. The nominal welding energy per unit length of the MAG process was 10.2 kJ/cm (travel speed 35 cm/min, voltage 27.8 V, current 215 A). The energy per unit length of the MIG process was 4.6 kJ/cm (travel speed 52 cm/min, voltage 24 V, current 175 A). The shielding gases used were 82% Ar/18% $CO_2$ for steel and 100% Ar for aluminum. The filler metals were EN ISO 14341-A-G4Si1 (S355N), G 89 4 M21 Mn4Ni2CrMo (S960QL), and EN ISO 18273:S Al 5356 (Al-6082 T6). Welding was conducted in a flat position, while the specimens were held in position by toggle clamps. Sheets of S960QL were welded at a pre-heating temperature of 100 °C. Representing macrographs and the hardness distribution (HV1) are shown in Figure 1.

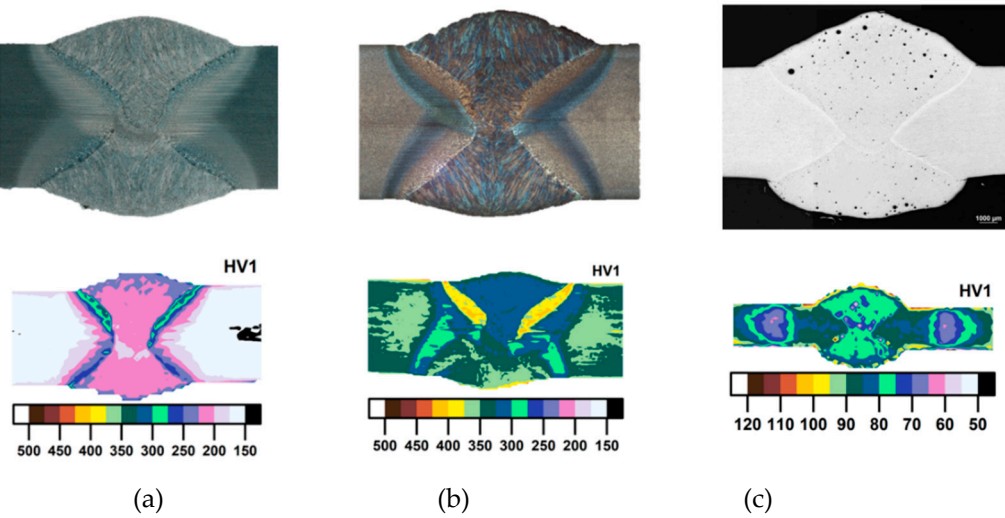

(a)                    (b)                    (c)

**Figure 1.** Macrographs of DV-butt welds. (**a**) S355N, (**b**) S960QL, (**c**) Al-6082 T6.

The hardness was measured with the ultrasonic compact impedance method (UCI), according to ASTM A1038. The equipment used was made by a BAQ UT200 hardness scanner (BAQ, Braunschweig, Germany). Typically, steels show heat affected zones with fine- and coarse-grained zones. The aluminum shows some pores and some small hot cracks in the vicinity of the fusion zone. However, the inner irregularities did not affect the fatigue failure, which occurred at the geometric notches at the weld toes. The weld quality was "B" according to ISO 5817 and ISO 10024. The welded steels showed an increase in hardness in the heat affected zone to approximately 300 HV1 (S355N) and 400 HV1 (S960QL). S960QL further showed an area of slight softening from approximately 320 HV1 (base metal) to 280 HV1 in the recrystallized zone. Al-6082 showed a softening of the base metal due to the dissolution of the precipitations. The hardness in these zones decreased to 60 HV1 compared to 85 HV in the base metal.

The 400 mm long welded plates were used to produce fatigue specimens by cutting and milling. The samples are shown in Figure 2. As the picture shows, the samples had different lengths, between 280 mm and 370 mm. However, this did not affect the fatigue test results, as the welding distortion was removed by means of a three-point bending device, resulting in nearly parallel surfaces in the area used for clamping. The plastic zones of the straightening process were located outside the fatigue

critical zone. Near-weld residual stresses were controlled before and after straightening to ensure that this had no effect on fatigue test results. For a more detailed explanation of the removal of distortion, see [18]. Furthermore, all samples were shot peened or clean blasted as a last preparation step before testing. All peening processes (test series 1 to 20) were performed air pressure-controlled, according to industrial standards at the facilities of OSK Kiefer GmbH in Oppurg, Germany. The peening of the samples led to a local increase in hardness. The near surface hardness of Al-6082 was increased to >125 HV0.03 by shot peening at coverage rates above 50%. In the case of S355N and S960QL, no significant increase in hardness was observed.

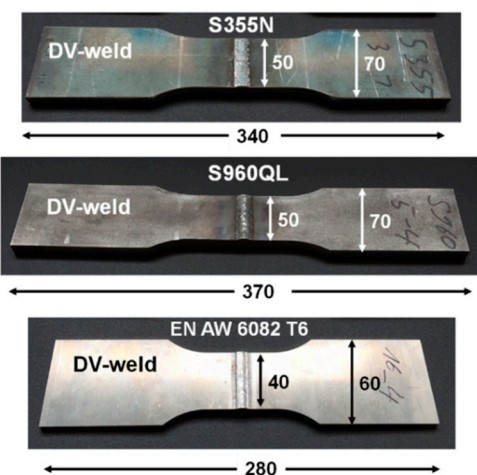

**Figure 2.** Fatigue test samples (DV-weld) shown in as-welded condition.

### 2.2. Residual Stress Measurements

Residual stress measurements were carried out at the residual stress laboratories at the Institute of Joining and Welding at TU Braunschweig. The methods used were X-ray diffraction (XRD), according to the $\sin^2$-$\Psi$ method [19], and incremental hole drilling (HD) [20]. Residual stress was determined from {211}-patterns of ferrite/martensite with the help of Cr-K$\alpha$-radiation at eleven $\Psi$-angles. Near-surface residual stresses were determined through incremental surface removal by means of electrolytic polishing and the hole drilling method. The collimator size was 2 mm for both steel and aluminum.

### 2.3. Fatigue Testing

Fatigue testing was conducted on servo-hydraulic test rigs (Schenck S400, Schenck, Germany; w + b250, Walter + Bai AG, Löhningen, Switzerland) at the Institute of Joining and Welding at TU Braunschweig. The test load was applied force-controlled uniaxial perpendicular to the weld. The stress ratio applied was $R = \sigma_{min}/\sigma_{max} = 0.1$. All tests were stopped at specimen fracture. An overview of all test series is given in Table 4. The fatigue strength of untreated specimens was used as reference to quantify the effect of peening. Reference fatigue strength is available for all three base metals used, S355N, S960QL, and Al-6082. Within the 20 test series, the peening media, the intensity, and the coverage were varied. The peening conditions were controlled according to state-of-the-art shot peening processes. It should be noted that a coverage below 98% is normally not used in shot peening (test series 14–16 and 18 and 19). These test series were included in the test matrix to investigate unfavorable peening conditions, e.g., during clean blasting.

Table 4 further shows the surface roughness of peened samples (MarSurf M 400 surface measuring instrument, Mahr GmbH, Göttingen, Germany). Glass beads generally create a lower roughness than steel shots. The highest roughness is generated by corundum due to its edged form. The high-strength steel S960QL shows lower roughness than S355N and Al-6082 due to its higher hardness.

**Table 4.** Overview about fatigue test series of peened DV-butt welds.

| Test Series | Base Metal | Peening Media | Intensity | Coverage | Surface Roughness $R_z$ |
|---|---|---|---|---|---|
| Reference | S355N | - (as welded) | - | - | |
| Reference | S960QL | - (as welded) | - | - | |
| Reference | Al-6082 | - (as welded) | - | - | |
| 1 | S355N | Steel shots | 0.3–0.4 mmA | 1 × 98% | 56.7 μm |
| 2 | S355N | Glass beads | 0.3–0.4 mmA | 1 × 98% | 38.4 μm |
| 3 | S355N | Corundum | 0.3–0.4 mmA | 1 × 98% | 71.9 μm |
| 4 | S960QL | Steel shots | 0.3–0.4 mmA | 1 × 98% | 25.8 μm |
| 5 | S960QL | Glass beads | 0.3–0.4 mmA | 1 × 98% | 30.1 μm |
| 6 | S960QL | Corundum | 0.3–0.4 mmA | 1 × 98% | 80.2 μm |
| 7 | Al-6082 | Steel shots | 0.3–0.4 mmA | 1 × 98% | 99.9 μm |
| 8 | Al-6082 | Glass beads | 0.3–0.4 mmA | 1 × 98% | 63.9 μm |
| 9 | Al-6082 | Corundum | 0.3–0.4 mmA | 1 × 98% | 107.5 μm |
| 10 | S355N | Steel shots | 0.3–0.4 mmN | 1 × 98% | 57.3 μm |
| 11 | S355N | Steel shots | 0.3–0.4 mmC | 1 × 98% | 61.3 μm |
| 12 | Al-6082 | Steel shots | 0.3–0.4 mmN | 1 × 98% | 83.3 μm |
| 13 | Al-6082 | Steel shots | 0.3–0.4 mmC | 1 × 98% | 96.5 μm |
| 14 | S355N | Steel shots | 0.3–0.4 mmA | 0.25 × 98% | 35.8 μm |
| 15 | S355N | Steel shots | 0.3–0.4 mmA | 0.50 × 98% | 37.1 μm |
| 16 | S355N | Steel shots | 0.3–0.4 mmA | 0.75 × 98% | 30.7 μm |
| 17 | S355N | Steel shots | 0.3–0.4 mmA | 2 × 98% | 63.8 μm |
| 18 | Al-6082 | Steel shots | 0.3–0.4 mmA | 0.50 × 98% | 91.9 μm |
| 19 | Al-6082 | Steel shots | 0.3–0.4 mmA | 0.75 × 98% | 88.2 μm |
| 20 | Al-6082 | Steel shots | 0.3–0.4 mmA | 2 × 98% | 96.5 μm |

Additional test series were prepared under less controlled conditions at industry workshops in a round-robin principle, Table 5. These samples, made from S355N, were provided to these companies for conduction of a "regular" cleaning process. The test series consisted of twelve specimens. The fatigue data was evaluated by linear regression without consideration of run-outs.

**Table 5.** Additional test series of industrially clean blasted specimen (round robin principle).

| Test Series | Base Metal | Peening Media | Shape | Participant |
|---|---|---|---|---|
| 21 | S355N | Steel shots | Edged | 1 |
| 22 | S355N | Cast steel shots | Edged | 2 |
| 23 | S355N | Cast steel shots | Edged | 3 |

## 3. Results

### 3.1. Residual Stresses in Conventional and Peened Specimens

The initial residual stresses were determined for all test series. The surface residual stresses in the as-welded condition (reference) are shown in Figure 3. The stresses were determined by means of XRD. Shown here are the mean values, as well as the absolute minimum and maximum values. The weld seam width is indicated by the hachured area. Note that diffraction patterns could not be obtained within the weld of S960QL and Al-6082 due to coarse grains. Residual stresses varied between 50 MPa and −150 MPa in specimens made from S355N and between 10 MPa and −200 MPa in S960QL. Aluminum welds show significantly lower residual stress with a mean close to zero, ranging from 70 MPa to −50 MPa. Low residual stresses can be explained by the low restraint condition transverse to the weld. This results in (relatively) free shrinkage and thus low residual stress. Compressive residual stress in the heat affected zones of the steels results from hindered volume increase due to phase transformation (austenite to ferrit/bainite/martensite) [21,22].

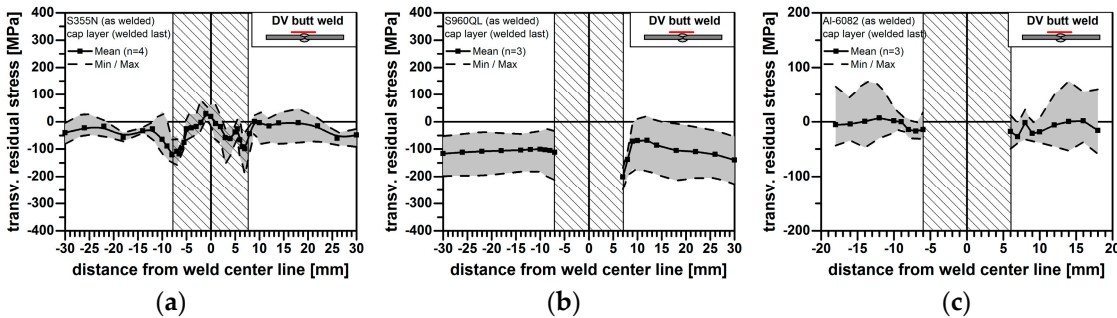

**Figure 3.** Residual stress at surface of DV-butt welds made from S355N (**a**), S960QL (**b**), and Al-6082 (**c**). Hachures indicate the weld seam width.

The residual stresses of the as-welded condition, shown in Figure 3, can normally be used as an indicator for residual stress effects on fatigue. However, this is not true for shot peened specimens. The reason for this is that the as-welded specimens show only a small residual stress gradient from surface into depth, while shot peening results in a relatively thin layer with distinctive residual stress gradients. The absolute value of the surface residual stress of shot peened samples at the surface is not a good measure for the beneficial compressive residual stress effect. It is recommended to determine the residual stress field within the first 1000 μm.

Figure 4 shows the residual stress conditions in all three investigated materials after clean blasting using different media: steel shots, glass beads, and corundum. The coverage and intensity were kept constant at 98% and 0.3–0.4 mmA. The results were obtained from the hole drilling method in the vicinity of the weld toe. It must be noted that a direct measurement at the weld toe by means of the hole drilling method is generally not possible due to geometric restraints (attachment of strain gauges). Glass beads and corundum generated comparable residual stress fields in S355N, with a minimum of approximately −440 MPa. The use of steel shots led to slightly higher compressive residual stresses and deeper penetration of the residual stress field. Not shown here is the influence of coverage. Its influence was investigated using S280 steel shots at a constant intensity of 0.3–0.4 mmA. The effect of coverage on the residual stress field in S355N was found to be low. The residual stress fields were comparable to the one shown in Figure 4a. Generally, the compressive residual stress magnitudes in S960QL were found to be higher than in S355N due to its higher yield strength. The peening media had a significant effect on the residual stress fields. The use of corundum led to the smallest penetration depth of the compressive residual stress field, with −550 MPa at a 120 μm depth. Glass beads and steel shots led to higher compressive residual stresses, −600 MPa at deeper layers (180 μm and 300 μm, respectively). The effects of coverage and intensity on the residual stress fields were also investigated here, with similar results to S355N (low effect).

Al-6082 showed generally lower compressive residual stresses due its lower yield strength. The influence of the shot media used was found to be relatively low. The highest compressive residual stresses were determined 200 μm below the surface. The magnitude was −200 MPa for all three media used. Not shown here is the influence of shot peening intensity. A higher intensity (0.3 mmA and 0.3 mmC) resulted in a deeper penetration of the compressive residual stress field than a low intensity (0.3 mmN). Furthermore, it was found that a low coverage of 75% and below resulted in lower compressive residual stresses than 98%, at less penetration of the residual stress field. However, 98% coverage and more (here up to 2 × 98%) did not further enhance the results.

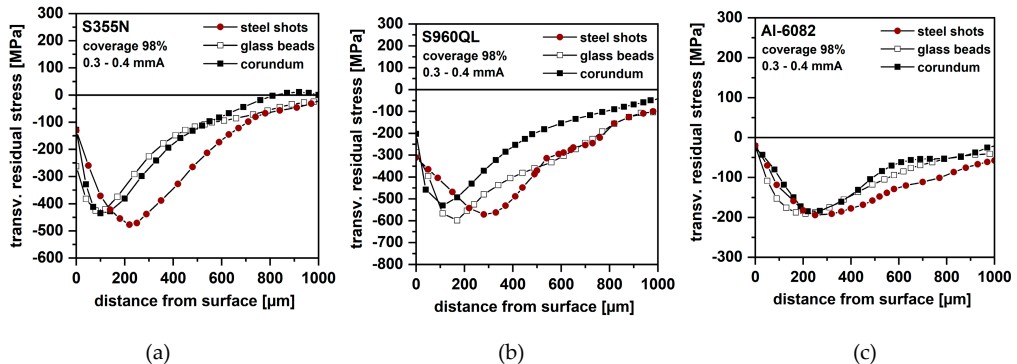

**Figure 4.** Residual stress in the surface layer obtained at the weld toes by means of hole drilling method. DV-butt welds made from S355N (**a**), S960QL (**b**), and Al-6082 (**c**).

## 3.2. Cyclic Stability of Shot Peening Residual Stress

Residual stress stability under cyclic loading was investigated using S355N samples. These were peened using steel shots at a normal intensity of 0.3–0.4 mmA, while the coverage was varied at 98% and 25%, Figure 5. Residual stresses were determined at the surface by means of X-ray diffraction. After assessing the initial residual stresses, the measurements were repeated after $N = 1, 10, \ldots 10,000$ load cycles. The fatigue loading was applied at a stress ratio of $R = 0.1$, with a maximum stress of 360 MPa. This was chosen to test the stability of the residual stresses under worst case conditions. It can be seen from the diagram that a coverage of 98% led to considerably more stable residual stress than a coverage of 25%. Although the initial conditions were comparable, stresses in the less covered sample were already relaxed at the first load cycle.

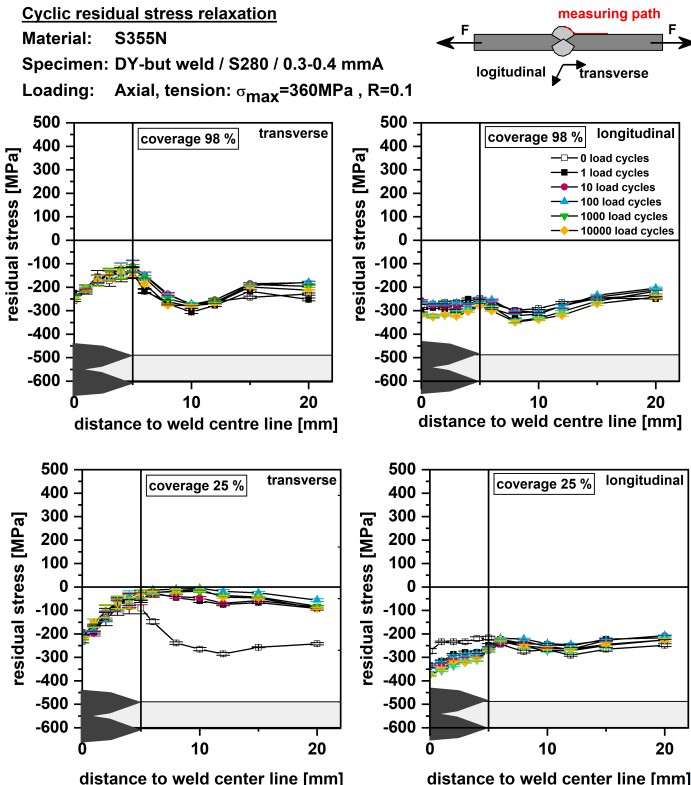

**Figure 5.** Residual stress relaxation in shot peened DV-butt welds. Use of steel shots at constant intensity. Variation of the coverage of 98% (**top**) and 25% (**bottom**). Cyclic loading at $R = 0.1$ with maximum stress of 360 MPa.

In the following, the in-depth residual stress of a pre-loaded S355N specimen were determined, as seen in Figure 6. The coverage was 98% using steel shots at 0.3–0.4 mmA. This test was conducted under the same loading conditions as described above. The stabilized residual stress was measured here using the hole drilling method and X-ray diffraction. It can be seen from the diagram that the entire compressive residual stress field is partly relaxed. However, the compressive residual stress of approximately −300 MPa remained effective.

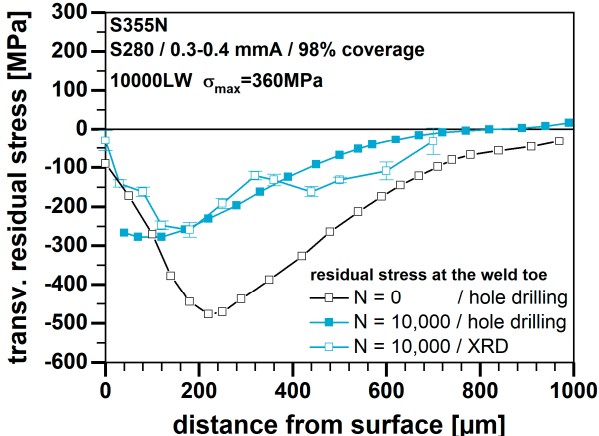

**Figure 6.** Change of residual stress depth profiles in shot peened (steel shots, intensity 0.3–0.4mmA, coverage of 98%) DV-butt welds under cyclic loading. Cyclic loading at *R* = 0.1 with maximum stress of 360 MPa.

### 3.3. Fatigue Test Results

During fatigue testing, it was noted that the different shot peening parameters had only very little effect on the resulting fatigue strength enhancement of S355N DV-butt welds. Because of this, all results from test series 1–3, 10, 11, and 14–17 (all test series using S355N) were treated as the same population and analyzed together, as seen in Figure 7. All these test results are summarized here under the simplified term "clean blasted", as no further distinction between the test series was made. The nominal fatigue strength range at 2 million load cycles was determined to Δσ = 254 MPa (50% POS—probability of survival). Additionally, shown here are the test series peened by industry partners, according to a round robin principle (participants 1 to 3). These results show similar fatigue strength to the "clean blasted" condition. Overall, the test data covers the finite life region between 105 and 106 well. The regression line was determined to *k* ≈ 10. Run-outs (specimen without failure) were stopped at 5 million load cycles.

The fatigue strength results from as-welded samples were determined in the IBESS-project and are given as a reference here [23]. Without going into details, the IBESS specimens were produced using the same welding parameters at the same facility. However, it must be pointed out that these tests were conducted at *R* = 0 and are not corrected to *R* = 0.1 due to its uncertain sensitivity to mean stress. The diagram further contains IIW FAT 90 (weld toe failure of untreated DV-butt welds) as a design fatigue strength. The as-welded fatigue strength was calculated to Δσ = 162 MPa at 2 million load cycles with *k* ≈ 4. The benefit of clean blasting (respectively shot peening) is most prominent at higher numbers of load cycles. The fatigue strength increase was determined to Δσ = 254 MPa/162 MPa = 156% at 2 million load cycles. Both peened and un-peened test series show comparable fatigue strengths below approximately 200,000 load cycles.

Figure 8 shows the test data from peened and un-peened specimens made from S960QL. Here, the peened test series are distinguished according to the peening media used (test series 4–6). Again, all results of peened specimens show similar fatigue strength between 260 MPa and 290 MPa at 2 million load cycles. Peening by means of steel shots and glass beads led to slightly better results,

which may be a consequence of the beneficial surface roughness. The inclination in the finite life region was determined to values between approximately $k = 6$ to 8. This can be explained by the higher yield strength of the base metal in comparison to S355N. The reference fatigue test data was taken from the IBESS-project as well. The fatigue strength as-welded was determined to $\Delta\sigma = 146$ MPa at 2 million load cycles with $k \approx 4$. In a direct comparison, shot peening led to an increase in the fatigue strength of approximately 180% to 200%. The fatigue strength increase is most prominent at high numbers of load cycles but also provable at higher stress amplitudes. This is explained by the high material strength and, thus, a higher residual stress stability. The differences in fatigue strength results between the three peened test series is most likely a result of the surface roughness. High strength materials usually show higher sensitivity to notch effects.

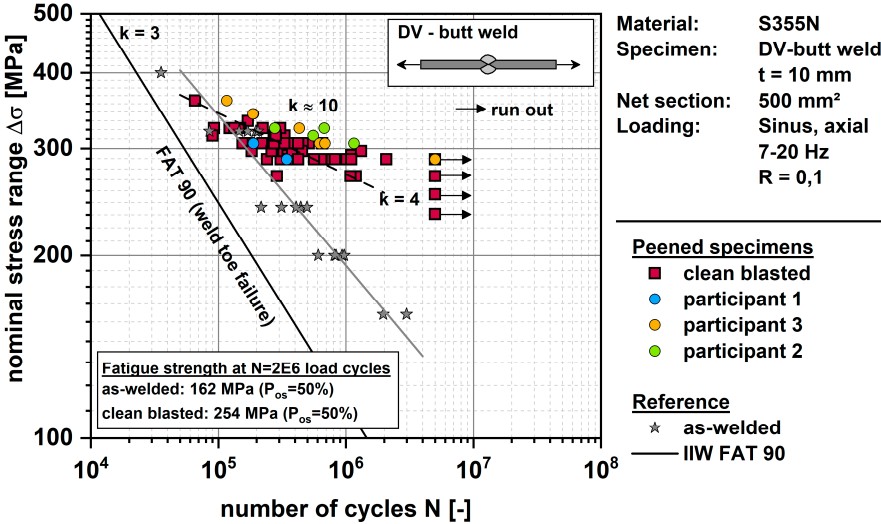

**Figure 7.** Fatigue test data of clean blasted DV-butt welds made from S355N. "Clean blasted" are results from test series 1–3, 10, 11, 14–17. Samples peened by participants 1 to 3 were clean blasted in industrial facilities and fatigue tested in Braunschweig (round robin principle).

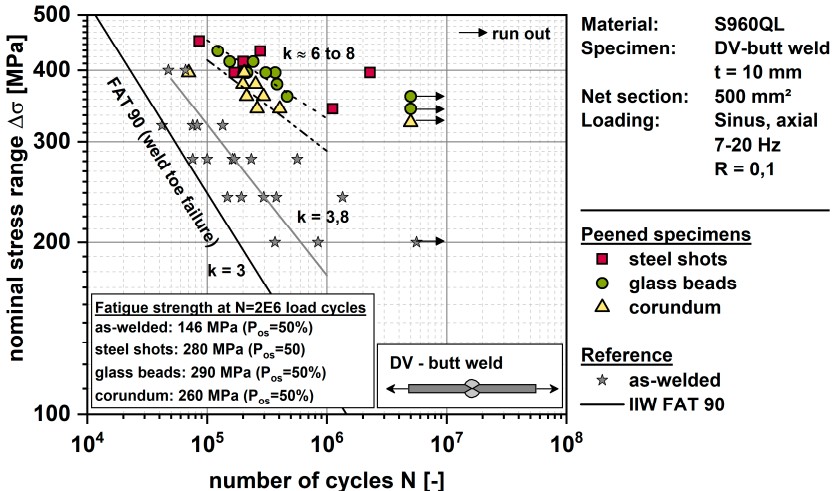

**Figure 8.** Fatigue test data of clean blasted DV-butt welds made from S960QL. Shown here are test series 4–9.

Figure 9 shows the results from tested Al-6082 specimens. Like results from S355N, the fatigue data of all aluminum test series was very similar and, hence, was analyzed as one group, without further distinction of the individual peening parameters. The group is therefore referred as "clean

blasted" (no peening parameters given). Here, run-outs were tested to 10 million load cycles. The determined fatigue strength at 2 million load cycles of clean blasted specimens was $\Delta\sigma = 94$ MPa. The regression line was determined to $k \approx 9$. Hence, the fatigue strength benefit due to post weld treatment is mostly effective at high numbers of load cycles. This is also explained by the relatively low yield strength of the material. The reference fatigue data (as-welded) were taken from a previous project [24]. These specimens were tested at $R = 0.1$, too. The fatigue strength at 2 million load cycles was determined to $\Delta\sigma = 58$ MPa ($k \approx 3$) and can be described by IIW's FAT 50 quite well. The increase in fatigue strength through clean blasting was determined to 162% at 2 million load cycles. However, the increase is limited again by the inclination of the finite life regression line.

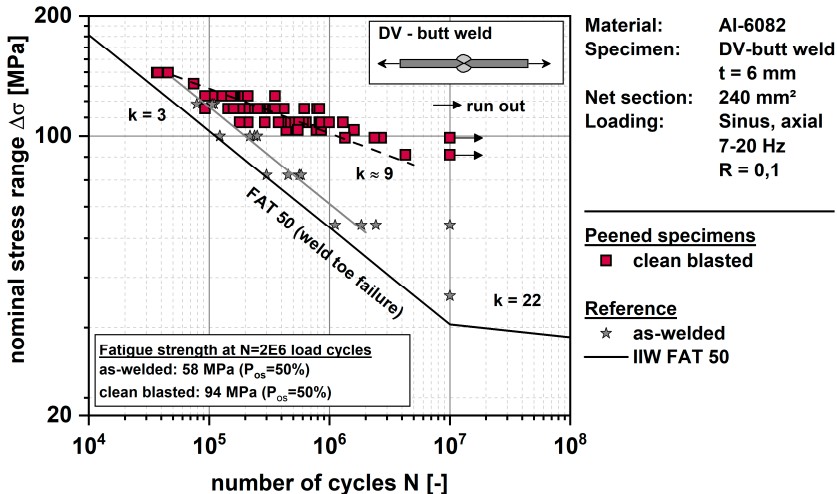

**Figure 9.** Fatigue test data of clean blasted DV-butt welds made from Al-6082. Shown here are test series 7–8, 12, 13 and 18–20 named as "clean blasted" without further distinction.

## 4. Discussion

Shot peening generates compressive residual stresses in all three materials. The surface residual stresses vary from test series to test series. Due to distinctive stress gradients, these values are not a good indicator for fatigue benefits through peening. The in-depth compressive residual stress field varies slightly in some of the test series. Steel shots with a coverage of at least 98% led to good penetration depth. However, the fatigue data did not reflect these differences of penetration depth. The peening process appears to be robust in terms of fatigue strength benefits. Nevertheless, the determination of cyclic residual stress relaxation indicates that a full coverage is beneficial for residual stress stability. This in good agreement with the literature [3]. Transferring these results to a possible utilization of clean blasting as a fatigue-enhancing post weld treatment method, full coverage should be always achieved. An explanation for the higher residual stress stability with increasing coverage is the more pronounced cold work hardening.

The fatigue data evaluated here (Figures 7–9) cannot directly be compared to IIW FAT-values because the data was not corrected for the stress ratio of $R = 0.5$ (IIW). Furthermore, the data was analyzed based on a 50% probability of survival. This study clearly shows the beneficial effect of shot peening on the fatigue strength of DV-butt welds made from a wide range of construction metals in direct comparison to as-welded butt welds. The fatigue strength enhancement was proven for industrially clean blasted specimens as well. Furthermore, the beneficial effect of peening on fatigue was proven, especially in the high cycle region of S-N curves, see Table 6. Tests using high strength steel S960QL additionally proved a fatigue strength enhancement at lower load cycles. The reason for this is the high yield strength of this material. Furthermore, the high strength of the base metal allows greater fatigue strength improvement, which may be explained by a combination of higher

compressive residual stress and high hardness. The high yield strength leads to a higher sensitivity to residual stress, resulting in a higher fatigue strength.

**Table 6.** Fatigue strength enhancement of the S-N curve in dependence of the base metal after shot peening.

| Base Metal | Fatigue Strength Enhancement at $N$ = 100,000 | at $N$ = 2,000,000 |
|---|---|---|
| S355N | - | ≈ 156% |
| S960QL | 125–140% | 180–200% |
| Al-6082 | ≈ 110% | ≈ 162% |

The peening process generated compressive residual stress in all test series. The fatigue data obtained does not indicate any advantage of peening media and/or parameters used for the soft metals S355N and Al-6082. The high strength steel showed some sensitivity to surface roughness induced by peening. The smallest fatigue strength improvement of S960QL was determined after the use of sharp edged corundum with a corresponding high roughness. This effect was not observed in the results from S355N or Al-6082, although the roughness was considerably higher after the use of corundum.

All results shown here were determined using butt welds with a relatively low stress concentration. General recommendations for bonus factors cannot be made yet as the data base, especially using weld joint types with higher stress concentration, such as cruciform joints, is missing. However, in a previous study, shot peening effects on the fatigue strength of longitudinal stiffeners made from S355N was investigated [25]. In contrast to the butt welds investigated here, the fatigue strength of longitudinal stiffeners was not also enhanced. The reason for this was found in the relaxation of compressive residual stresses at certain load amplitudes. As a result, the fatigue strength of as-welded and shot peened samples was comparable at 1 million load cycles and below. A benefit was proven only at load stress amplitudes of 60 MPa and less ($R = -1$), due to a higher residual stress stability. These results show the effect of stress concentration at the weld on the success of shot peening as a post-weld treatment method. This should be studied further in future.

## 5. Conclusions

Fatigue strength can be significantly improved by shot peening. Round robin tests with S355N specimens have further proven the high potential of regular industrial clean blasting processes to achieve similar results. The effect of shot peening on fatigue strength increases with increasing yield strength of the base metal. Coverage should be controlled and chosen to be at least 98% to improve residual stress stability under mechanical loading. The peening media affects the surface roughness. In the case of high strength steels, roughness becomes important, as an increase in roughness lowers the fatigue strength. Stress concentration may influence the residual stress stability and thus the fatigue strength. This should be further investigated.

It is recommended to qualify industrial clean blasting processes with the help of the Almen test, roughness measurements and, as far as possible, residual stress measurements and component fatigue tests. By such a qualification of the clean blasting, designers may use bonus factors for the fatigue design of welded components.

**Author Contributions:** Conceptualization, T.N.-P. and J.H.; methodology, T.N.-P. and J.H.; validation, T.N.-P.; investigation, H.E. and J.H.; resources, K.D.; data curation, H.E. and J.H.; writing—original draft preparation, J.H.; writing—review and editing, T.N.-P..; visualization, J.H.; supervision, T.N.-P.; project administration, T.N.-P.; funding acquisition, T.N.-P. and K.D.

**Funding:** This IGF project IGF-Nr.: 18.985 N of the "Forschungsvereinigung Schweißen und verwandte Verfahren e. V. des DVS" was supported via AiF within the program for promoting the Industrial Collective Research (IGF) of the German Ministry of Economic Affairs and Energy (BMWi), based on a resolution of the German Parliament.

**Acknowledgments:** The authors thank OSK Kiefer, Oppurg, Germany, for the peening of samples. We acknowledge support by the Open Access Publication Funds of the Technische Universität Braunschweig.

**Conflicts of Interest:** The authors declare no conflict of interest.

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
