# Peer review of "Fatigue Strength Enhancement of Butt Welds by Means of Shot Peening and Clean Blasting"

_metals, doi:10.3390/met9070744_

Round 1
Reviewer 1 Report
The paper deals with Fatigue strength enhancement of butt welds by means of shot peening and clean blasting. The manuscript is generally well organized and written. design method is appropriate and conclusion are supported by results.
The reviewer only suggests modifying the scale of y-coordinate of figures 7 and 8 in order to better read the different tests.
Author Response
The reviewer only suggests modifying the scale of y-coordinate of figures 7 and 8 in order to better read the different tests.
Thank you very much. The scale of the y-axis is adjusted from 100 to 500 MPa.

Reviewer 2 Report
In this work the authors analyze the effect of shot peening and clean blasting on Fatigue strength enhancement of butt welds. The research appears to be efficiently done and appropriately reported, however the standard of English must be substantially improved. Nevertheless, there some questions and corrections that must be answered to improve and complete the document.
Line 80- Please change the word “sur-face” to “surface”.
Line 87- The sentence is grammatically incorrect, please correct it.
Line 111 – The authors present the expression “0.25x98%”, however they never explained objectively the meaning of “0.25”. Please, at least, in the first time that you write this expression must explain the meaning of it.
“Chapter 2 – Experimental work” is very incomplete, there are many steps and details that must be included in the present paper. Please, be attention to follow comments and questions:
- Lines 126-129: the authors write that they welded plates with, approximately, 400 mm length, however, the information of thickness and width must be included. In other hand, they do not indicate any information about the experimental test: which are the welding parameters (current, voltage, arc travel speed, length of rrc, work and travel angle, wire materials, shielding gas…)? What kind of butt weld geometry was used? square? single V? double V? Did you used back weld? How was the clamping of the plates? Did they use a robot? Which one? Which welding machine did they use? Please, present an image with the experimental set-up.
- Lines 134-138: How did you measure the hardness? Equipment used? Regions of sample where the measurements were done? Did you use any software to treat the data of measurements? Which one?
- Lines 148-149: What was the machine where the samples were peened? Brand and model?
- Lines 156-160: Please, detailed the residual stress measurement procedure?
- Line 163: The authors describe the fatigue test indicating “The test load was applied …”. However, this description do not give enough information to understand the experimental set-up. I suggest that an image of experimental set-up could be very useful.
- Line 164: the equation is incomplete, probably, some information had lost when was created the pdf. Please, be attention to this detail.
- Line 170: Please verify the sentence. There is a mistake in the grammatical construction.
- Lines 179-180: This is not a sentence that you should use in scientific paper. Please, remove the sentence or rewrite it in a more scientific style.
- Lines 180-181: The sentence “The participants … twelve specimens.”. Is very confuse, please rewrite it in a way that could be more intelligible.
Lines 226 and 305: Please, change the word “deter-mined” to “determined”.
Lines 228: Please, change the word “penetra-tion” to “penetration”.
Line 324: Please change “This in …” to “This is in …”. In tis sentence, the authors claimed that the full coverage is beneficial for residual stress stability is in agreement with the literature, however, they don’t indicate any reference to confirm this statement. Please, indicate one or two references to prove the statement.
Line 331: Please verify the sentence of this line and rewrite it.
Line 338: the authors write “ … sensitivity to residual stress m resulting…”. Please verify if there are any mistake or any problem happened during the document conversion of word to pdf.
Author Response
Thank you for your review. Your comments have helped to improve the paper.
Line 80- Please change the word “sur-face” to “surface”.
Thank you very much.
Line 87- The sentence is grammatically incorrect, please correct it.
Thank you very much. The sentence is re-structured.
Line 111 – The authors present the expression “0.25x98%”, however they never explained objectively the meaning of “0.25”. Please, at least, in the first time that you write this expression must explain the meaning of it.
This is explained in the text, when discussing “coverage” the first time. Please see lines 81ff
“Chapter 2 – Experimental work” is very incomplete, there are many steps and details that must be included in the present paper. Please, be attention to follow comments and questions:
- Lines 126-129: the authors write that they welded plates with, approximately, 400 mm length, however, the information of thickness and width must be included. In other hand, they do not indicate any information about the experimental test: which are the welding parameters (current, voltage, arc travel speed, length of rrc, work and travel angle, wire materials, shielding gas…)? What kind of butt weld geometry was used? square? single V? double V? Did you used back weld? How was the clamping of the plates? Did they use a robot? Which one? Which welding machine did they use? Please, present an image with the experimental set-up.
Thank you very much. These details are now given in line 128ff. The width of samples is shown in Figure 2. An additional figure of the setup is not added. The reason is that this paper focuses on the shot peening effects on fatigue of weldments, not the welding experiments.
- Lines 134-138: How did you measure the hardness? Equipment used? Regions of sample where the measurements were done? Did you use any software to treat the data of measurements? Which one?
The hardness was measured with the Ultrasonic Compact Impedance method (UCI) according to ASTM A1038. The equipment used was a BAQ UT200 hardness scanner (BAQ, Braunschweig, Germany). The plot was made with help of Origin without any post-processing of the data. Line 137ff
What was the machine where the samples were peened? Brand and model?
The peening was conducted at industrial facilities. The brand and model of the machines used are not available. However, all essential peening parameters are given in line 158ff. These parameters ensure that the process is repeatable also by others. The device used for peening used pressured air.
- Lines 156-160: Please, detailed the residual stress measurement procedure?
Ok, see line 167ff
- Line 163: The authors describe the fatigue test indicating “The test load was applied …”. However, this description do not give enough information to understand the experimental set-up. I suggest that an image of experimental set-up could be very useful.
Thank you very much. This is described in L174ff. The test load was applied force-controlled uniaxial perpendicular to the weld. This is a standard test for the determination of fatigue strength of welds.
- Line 164: the equation is incomplete, probably, some information had lost when was created the pdf. Please, be attention to this detail.
Thanks, there was a mistake
- Line 170: Please verify the sentence. There is a mistake in the grammatical construction.
Ok, the sentence was re-worked.
- Lines 179-180: This is not a sentence that you should use in scientific paper. Please, remove the sentence or rewrite it in a more scientific style.
Ok, the sentence was deleted
- Lines 180-181: The sentence “The participants … twelve specimens.”. Is very confuse, please rewrite it in a way that could be more intelligible.
Ok, the sentence was re-worked.
Lines 226 and 305: Please, change the word “deter-mined” to “determined”.
Thank you very much.
Lines 228: Please, change the word “penetra-tion” to “penetration”.
Thank you very much.
Line 324: Please change “This in …” to “This is in …”. In tis sentence, the authors claimed that the full coverage is beneficial for residual stress stability is in agreement with the literature, however, they don’t indicate any reference to confirm this statement. Please, indicate one or two references to prove the statement.
Thank you very much. Two references are added
Line 331: Please verify the sentence of this line and rewrite it.
Line 338: the authors write “ … sensitivity to residual stress m resulting…”. Please verify if there are any mistake or any problem happened during the document conversion of word to pdf.
Thank you very much. The sentence is corrected.

Reviewer 3 Report
Paper is very interesting and well-written. Just few minor revisions are suggested to improve the readability of the paper.
Introduction, page 1, line 30: reviewer suggests considering the following paper (not-mandatory) on the post-weld treatment: Sepe, R., Armentani, E., Lamanna, G., Caputo, F., Evaluation by FEM of the influence of the preheating and post-heating treatments on residual stresses in welding (2015) Key Engineering Materials, 627, pp. 93-96.
Page 2, lines 52-52. Authors should justify the statement “Unfortunately, benefits of shot peening on fatigue strength 52 can not be used in design of welded structures yet”. In addition, reviewer suggests to replace can not with “cannot”.
Page 5, line 164. Check the definition of R parameter, please.
Page 7, “Results” section. Reviewer suggests specifying the Yield stress of the specimens during the residual stress analysis.
Figure 6 caption. Check “Us”.
Author Response
Introduction, page 1, line 30: reviewer suggests considering the following paper (not-mandatory) on the post-weld treatment: Sepe, R., Armentani, E., Lamanna, G., Caputo, F., Evaluation by FEM of the influence of the preheating and post-heating treatments on residual stresses in welding (2015) Key Engineering Materials, 627, pp. 93-96.
Thank you very much for your comments. This reference was not added to this section mainly because the shot peening improvement is discussed in the context of design guidelines for fatigue design. However, the authors are aware of the wide field of literature on research explaining the individual effects of different improvement methods.
Page 2, lines 52-52. Authors should justify the statement “Unfortunately, benefits of shot peening on fatigue strength 52 can not be used in design of welded structures yet”. In addition, reviewer suggests to replace can not with “cannot”.
Thanks, this was corrected.
Page 5, line 164. Check the definition of R parameter, please.
Thanks, this was corrected.
Page 7, “Results” section. Reviewer suggests specifying the Yield stress of the specimens during the residual stress analysis.
The yield strength of base metals used are given in Tab. 1. For this reason, this is not repeated here. However, the determined residual stress values range well below the yield limits, as it would be expected.
Figure 6 caption. Check “Us”.
Thanks, this was corrected.

Round 2
Reviewer 2 Report
The second version of manuscript improved significantly when compared with first version. So, in my opinion the manuscript can be accepted for publication.